# From Liminal Labor to Decent Work: A Human-Centered Perspective on Sustainable Tourism Employment

**Dimitri Ioannides** [1,*] , **Szilvia Gyimóthy** [2] **and Laura James** [3]

1    European Tourism Research Institute, Mid-Sweden University, 83125 Östersund, Sweden
2    Department of Marketing Copenhagen Business School, 2000 Frederiksberg, Denmark; sgy.marktg@cbs.dk
3    Department of Culture and Learning, Aalborg University, 9220 Aalborg, Denmark; leja@hum.aau.dk
*    Correspondence: dimitri.ioannides@miun.se

**Abstract:** In its sustainable tourism agenda for 2030, the UN World Tourism Organization has embraced three United Nations Sustainable Development Goals. One of these, specifically SDG 8, highlights the need to pursue decent work and growth. Nevertheless, despite the growing recognition of this target and although there is a growing number of writings lamenting the precarity characterizing many tourism-related jobs, the topic of tourism-related work continues to receive sparse attention in the considerable volume of academic literature on tourism and sustainability. This paper attempts to redress this neglect. First, by providing a review of extant studies on tourism labor, we seek to explain why this research lacuna continues to exist. We then examine organizational and technological aspects of tourism governance, which hinder attempts to establish decent work and improve dignity in the tourism industry worldwide. By acknowledging the volatile and liminal status of tourism work and future labor market prospects, we arrive at the following question: what should sustainable tourism work look like? This leads us to suggest that the development of a human-centered research agenda, which focuses on workers' agency and resources, offers a promising research avenue for expanding on the tourism and sustainability research agenda.

**Keywords:** tourism work; worker agency; precarity; sustainability; job crafting; human-centered agenda

## 1. Setting the Scene

The recently published UN policy brief *COVID-19 and Transforming Tourism* [1] paints a dire picture of the pandemic's negative impacts on tourism worldwide. It highlights a catastrophic loss of 100 million jobs directly attributed to tourism and indicates that the most vulnerable persons in the sector, many of whom are precariously employed [2], are women, young persons and migrant workers. The policy brief also stresses how the pandemic has stalled progress towards achieving several sustainable development goals including SDG 1 (on poverty reduction), SDG 5 (on gender equality) and SDG 8 (on decent work and economic growth). The latter part of the document argues that "the crisis is also an unprecedented opportunity to transform the relationship of tourism with nature, climate and the economy" (p. 4) and includes broad, albeit vague, suggestions on ways to improve the working conditions for millions of workers. Ultimately, the UN policy brief expresses the lofty ambition that in the aftermath of the pandemic it behooves us to reset the global tourism sector on a more sustainable, resilient and inclusive path.

As noble as this ambition is, it ignores the fact that most stakeholders—including agencies at all levels of governance as well as businesses—have a superficial perspective on sustainable tourism, predominantly focusing on the sector's perpetual growth while treating the other dimensions of sustainable development (environmental protection and the promotion of social justice and equity) as secondary concerns [3]. Developers or tourism promotion organizations may see environmental protection as a way of reinforcing the image of a particular destination in order to increase its marketability [4,5]. This implies that their motives for promoting this goal are rarely altruistic. Policymakers regularly

tout tourism as a means of generating economic growth and jobs while overlooking the conundrum arising from the fact that a particular locality may lack a labor force large and diverse enough to fill all created positions or that the conditions for many workers in the sector are substandard given the high degree of precarity characterizing many such jobs [6]. Along these lines, research as to how tourism work and workers fit into the overall sustainability discourse, especially when it comes to the key dimension of social justice and equity, is surprisingly scarce [7]. Bianchi and de Man [8] view this oversight as a direct outcome of the fact that the prevailing pro-growth perception of tourism as a driver of wealth creation contradicts the pessimistic narratives relating to inequalities and outright exploitation arising from the sector. Meanwhile, Wakefield [9] is troubled that liberal development agendas treat the human subject as separate from the environment while objectifying the individual as "variously a nugget of labor power or a docile subject to be shaped and molded by external forces" (p. 27).

Several observers have commented on the limited attention to tourism work and workers in academic scholarship [10–13]. Baum et al. [13] have pinpointed that many such investigations "suffer from piecemeal approaches at topic, analytical, theoretical and methods levels" (p. 1). Consequently, they recommended that, by adopting a taxonomy for the tourism workforce, we can develop a more comprehensive understanding as to its structure and characteristics. Meanwhile, Baum et al. [7] have highlighted the persistent neglect of employment-related issues within the overall "sustainable tourism narrative" (p. 1) while seeking to link this topic to the United Nations' 2030 Agenda for Sustainable Development [14]. They wished to flesh out the connections between sustainability and several aspects of tourism-related work and workers while examining how these relate to the principles of sustainable human resource management. Their ambition was to suggest that matters relating to employment and the labor force must occupy the center-stage of discussion on sustainability. However, they pessimistically concluded that despite the widespread understanding that working conditions in the tourism sector are overwhelmingly poor, both the tourism industry and academics persistently fail to address ways to improve the situation (see also [15]). In a later commentary, Baum et al. [16], (p. 252) expressed skepticism that "anything on the horizon within tourism and its wider socio-economic, technological and environmental context" will transform how tourism work is performed by the end of the 21st century. This opinion contradicts Wirtz et al. [17], who believe that new technologies such as robotics will have a major effect on service industries. Instead, Baum et al. warn that new practices such as the advancement of artificial intelligence and robots could undermine the very goal of improving the quality of tourism jobs.

Inspired by research on tourism labor [7,8,15,18], we argue that in order to move the discussion on sustainable development forward in accordance with the overall ambition of this Special Issue, we must embrace the topic of tourism work and workers and try to better understand what tourism employment in the context of sustainability means. Among the broad questions that emerge are: How do we reconcile the fact that a high proportion of jobs in the sector can be described as precarious with the need to match calls for creating decent jobs according to the UN SDGs? What obstacles (societal, institutional, sectoral) hinder this objective from happening? What does sustainable employment in the sector actually mean and how do we achieve this? Certainly, we do not aim to answer all of these quite broad questions within a single article. Rather, more realistically, we wish to unravel certain key issues with the hope that these will enable us to propose a research agenda for the future.

From the outset, we acknowledge that a critical overview of the precarious nature of tourism work is not constructive on its own if we wish to move the dialogue on this matter further. Critiques are useful in order to understand why a problem exists in the first place. However, they offer little as to how to overcome this situation. Thus, as a first step, we warn our readers that we do not portray tourism workers as voiceless, passive victims and marginalized subjects of production. Instead, we attempt to flesh out the heterogeneity

and differential power dynamics that characterize various tourism workers. By casting light on the broader socio-spatial context of tourism labor and job crafting, we contribute to existing discussions of individual agency, motives and choices, which help better position tourism workers in the overall sustainability debate [15,19].

## 2. Literature Background

### 2.1. The Precarious Nature of Tourism Work and the Forces behind This

The rise of precariousness in the global work force has been well-documented in recent years. In general, precarity suggests "a state defined by a lack of security and predictability" while, specifically, "precarious work is characterized by employment that is irregular and insecure" [15], (p. 1011). In such a scenario, the risk is increasingly transferred from the employer to the workers while employer obligations to provide benefits (contributing towards the employees' social security, pension plans or, where relevant, health insurance) are substantially reduced. Various jobs fit this label, including (but not limited to): temporary agency-based work; casual work (e.g., seasonal and/or part-time; many types of home-based employment or working for a platform-based employer such as UBER or Deliveroo. Lambert and Herod [2] argue that the precarity witnessed in the global labor force results directly from the proliferation of neoliberal policies, including widespread deregulation, over the last three decades. According to Herod [6], (p. 81), the International Labor Union (ILO) highlighted that in 2015 just "about 25% of workers worldwide have any kind of stable employment relationship." Herod argues that the rest of the global labor force is employed on a contingent basis. For example, some individuals work informally for a family-run business or are seasonally involved in jobs such as fruit picking or operating a ski lift.

In certain countries, especially within the Global North, more and more individuals choose a part-time position for the purposes of flexibility. Often, they base their decision on their lifestyle (e.g., retirees working part-time to supplement their pension or persons who feel they earn enough by working as independent contractors from home). Conversely, most precarious workers in advanced economies but mostly in the Global South have limited choices when it comes to their employment conditions. Often, they are hostage to their employers' whims and unable to negotiate a better working contract.

Robinson et al. [15] insist that while in other industries the accentuation of labor precarity is fairly recent, in the case of tourism and hospitality, many of the characteristics of precariousness have been around for decades. It is hardly surprising, therefore, that several authors have discussed labor precarity within this sector [20–23]. Rydzik and Anitha [24] highlight the precarious working conditions of many migrants in the UK tourism industry who are especially vulnerable either because they are unaware of their rights or because of weak language skills. Meanwhile, employers exploit these individuals' weak bargaining power, leading to situations of enhanced precarity. At the same time, the experiences of these workers are shaped by complex intersections of gender, class and ethnicity, and associated divisions of labor (McDowell, Batnitzky and Dyer, 2009; Lugosi et al. 2016).

Winchenbach et al. [19] describe how, in pursuing profit maximization, tourism and hospitality companies regularly exploit employees by overworking and underpaying them while rarely providing opportunities for promotion. Baum [21,25] has repeatedly critiqued the tendency to look down on tourism and hospitality jobs, especially as these are often the last resort for those desperately searching for employment. To illustrate the poor status of tourism-related work, Baum [21] refers to George Orwell's experiences when the (then aspiring) author worked as a plongeur, a dishwasher in a Parisian hotel in the 1930s in order to make ends meet.

Several observers overwhelmingly associate tourism with low-wage jobs, which are often part-time, temporary and/or seasonal [26]. Others [10] have argued that many of the low-skill jobs (e.g., dishwashing or hotel room cleaning) are filled by women and/or immigrants, especially from the Global South. In her ground-breaking ethnographic study

*Nickel and Dimed: On (Not) Getting by in America*, Barbara Ehrenreich [20] vividly illustrated how, in the US, women with limited opportunities face extreme hardship when working as waitresses or hotel housekeepers for lowly hourly wages with no benefits. Many have multiple jobs to make ends meet. The same precarious labor conditions apply to urban-based illegal immigrants who are traditionally excluded from standard employment opportunities. Van Doorn and his colleagues [27] demonstrate that the casual jobs in the platform economy (offered by gig companies like Uber, Helpling or Deliveroo) are primarily performed by migrant workers. Hospitality and platform-based gig workers are equally vulnerable and disposable, owing to limited regulations, labor and wage protection and high risks of discrimination characterizing these job opportunities [28]. Meanwhile, Winchenbach et al. [19] argue that, in sectors like tourism, the absence "of dignity and respect, unequal power relations and poor working conditions create a sense of alienation and mistrust, negatively affecting the success of the business as well as workers and local communities" (p. 1029).

Shaw and Williams [29] have demonstrated how the limited skills associated with numerous jobs on the lower end of the spectrum in the tourism sector cause wage suppression. This, in turn, causes high labor turnover, a problem that is compounded since employers treat their workers as costs rather than long-term resources and see them as highly substitutable [8]. Further, because tourism is often weakly unionized, partly because of the sector's highly fragmented nature and employers' overwhelming antipathy towards organized labor, this significantly reduces the workers' necessary bargaining power for improving their conditions [30].

As in many other sectors, several forces reinforce the precarity of tourism-related work [6]. An important factor is the high degree of numerical flexibility characterizing several aspects of the tourism and hospitality sector [29]. In the so-called post-Fordist era, many tourism and hospitality firms adopt such an approach, allowing them to quickly increase the numbers of workers when necessary (for instance when demand levels are high) while decreasing them in times of slowdown [31]. Robinson et al. [15], (p. 1011) highlight that "the numerical and functional flexibility afforded to employers, and the ability to reduce the payroll at a moment's notice contributes to nimble firms not constrained by the permanency of a standing workforce." Head and Lucas [32] noted that half the accommodation businesses in London employ part-timers, a trend that has increased over the years because of the growing tendency to use agency-based staff to cope with unpredictable variations in room occupancy. Meanwhile, Lee et al. [22] discuss how, in pursuing foreign direct investment, the Seychelles have enabled multinationals to erode the power of locally owned businesses. In turn, this has reduced the protections for local workers who are now more susceptible to exploitation.

Robinson et al. [15] underscore tourism's guilt in accentuating the precarity commonly associated with tourism-related work. In their mind, it is precisely this situation that "contributes to deep social cleavages and economic inequalities", which in a vicious circle reinforces the "precarious nature of work itself" (p. 1009). They believe that this situation is unsustainable in the long-run and stress the necessity to incorporate people and especially tourism workers, in discussions relating to sustainability. These authors argue that this can only be achieved once social-related issues are elevated to the same status as the environmental and economic growth concerns, which traditionally have dominated the sustainability debate (see also [7]).

*2.2. Where Do Tourism Work and Workers Fit within the Sustainability Debate*

Given the precarity of many tourism-related jobs it is surprising that the topic has received sparse attention in the hundreds, if not thousands, of academic publications and scientific reports on sustainability and tourism published since the late 1980s. Observers have frequently noted the scholarly neglect of the social side of sustainability, which includes elements relating to work and workers [15,33–35]. Baum [7] (p. 873) stresses the need to overcome the neglect of "the tourism workforce and associated employment issues

from a sustainability perspective", arguing that "workforce and employment issues in tourism cannot be interpreted without reference to the wider, social, cultural and economic context within which they are identified". He is puzzled by the neglect of tourism employment and matters relating to work quality in discussions on sustainable development, especially considering that, at the level of the firm, many companies have shifted their human resource management strategies in the spirit of corporate social responsibility (CSR) to improve their hiring practices and the working conditions for their employees. Similarly, Robinson et al. [15] maintain that tourism workers regularly engage in sustainability-related activities in day-to-day operations (e.g., recycling and encouraging guests to reuse sheets and towels as a water-saving measure) and yet they are inadequately treated in theoretical or policy-driven discussions on sustainability.

Several reasons explain this neglect. A key concern relates to tourism's fuzzy industrial classification since it is hard to statistically pinpoint what constitutes the tourism sector or, indeed, a tourism job [25,36]. Yet another reason for avoiding labor is the way that sustainability issues have been framed in relation to tourism in general. Typically, these are explored in terms of the contradiction between preserving the natural environment while promoting economic growth [37]. Meanwhile, both the resource-based and the community-based perspectives of sustainability identified by Saarinen [38] fail to address the tourism workforce adequately. This largely relates to the fact that it is hard to identify the status of what is often a highly mobile workforce in these perspectives, which focus predominantly on local conditions in any given community.

To begin with, the resource-based approach focuses on the depletion of resources in a certain area because of, for instance, intensified agricultural or industrial production. In a mass tourism context, the resource-based view concentrates on the exploitation of assets (natural environment, cultural heritage, local capital) involved in producing tourism. Thus, in this situation, the emphasis is primarily on damage limitation. This can be achieved by assessing the level of disturbance inflicted on local resources, in order to establish (or extend) the limits of local growth [39]. Thus, if, for example, we examine carrying capacity frameworks applied to tourism destinations, these only consider employees in terms of their volumes (e.g., the proportion of seasonal workers in relation to the host population) [40] while ignoring their working conditions and personal circumstances. Obviously, therefore, the resource-based perspective treats tourism workers in a utilitarian way while concurrently alienating them as a predominantly migrant labor force belonging to global tourism production systems (Robinson et al. 2014) [12] or, more optimistically, perceiving them as potential sources of innovation. Williams and Shaw [41] discuss how global migration flows of workers might remove several important obstacles to tourism innovation. They argue that skilled migrant workers are, therefore, more likely to accept new ways of doing things. This in turn makes it easier for companies to embrace innovative practices. Moreover, since migrants, especially from the Global South, are usually paid lower wages than local workers, this frees up capital to invest in various technologies.

In contrast to the resource-based view, the community-based approach to sustainability [42] stresses the importance of social capital and empowerment/involvement of the host community in tourism development [38]. Several academics have undertaken research examining the intersections of tourism with residents' everyday life, and the operations of local entrepreneurs [43]. Studies have attended to the power asymmetries within local communities, pinpointing the differences in how local actors benefit from tourism [44], but the community-based approach also avoids an explicit consideration of tourism workers as a part of the local. Rather, it sees the migrant—often seasonal—workforce as external and segregated from the host population, which in the sustainability lens often tends to be seen as underprivileged [15,45,46]. Due to the considerable ambiguity that characterizes the affiliation of tourist workers in terms of the localities where they are employed, these individuals often remain under the radar of most researchers. This means that because in numerous destinations the tourist workers are rarely long-term residents (e.g., guest workers in the destination on a temporary/seasonal basis), scholars who examine sustain-

ability from the point of host societies tend to overlook them. Salazar [18] highlights this problematic issue by inquiring how the sustainability of a particular place, which suffers from labor shortages, is affected by the influx, not only of tourists, but also of a highly mobile workforce during the peak tourism season. When tourism workers are considered in relation to sustainability, it is often in terms of the impacts these individuals have on destinations rather than the employment conditions the migrants themselves face [47].

In sum, we believe that the tendency in much of the tourism sustainability literature to focus on how local people, local resources and local culture become hostage to the whims and fancies of powerful non-local (outsider) interests obfuscates the precariousness, especially of non-resident groups in various communities. Thus, tourism workers remain exogenous to local ecosystems and since they often lack the permanency of territorial belonging that characterizes other groups (i.e., long-term local residents), they are rendered "homeless" in different sustainability discussions.

This leads to the following question: What should sustainable tourism work look like? Baum et al. [7] highlight SDG 8 of the United Nations' 2030 Agenda for Sustainable Development as particularly relevant to discussions revolving around tourism-related work and workers since it focuses on the concept of "decent work" as championed by the International Labor Organization (ILO) from the late 1990s onwards. Decent work operationalizes the notion of economic sustainability in relation to labor markets and is now frequently invoked as the critical element in the conceptualization of sustainability and work [7,19,25]. The ILO's decent work pillars comprise: rights at work (freedom of association and the right to collective bargaining, abolition of child labor, elimination of forced labor, elimination of discrimination in employment); fostering employment (policies and strategies to achieve full employment with appropriate pay, but including all forms of work that contribute to society including unpaid and informal work); social protection (prevention of oppressive and unsafe workplace conditions as well as social security and paid holidays); and social dialogue (consultation, negotiation and agreements between workers and employers) [48,49].

Thus, as Winchenbach et al. [19] argue, sustainable tourism work implies more than mere job creation and involves several labor-related policy traits, including: the need to pay fair wages; provide safe spaces of employment and protection safeguards for employees and their families; strive for equal opportunities regardless of gender or race; enabling workers to fight for their rights without fear of recrimination; and offering opportunities for upward career mobility. Scheyvens [50] argues that decent work necessitates, among others, that women in the sector can work in a safe, threat-free environment. Unfortunately, despite these ambitions, much of the work performed in tourism fails the standard of what "decent" actually means, precisely because of the ongoing tendency to neglect "workforce and workplace considerations in the growing volume of debate relating to sustainable tourism" [7], (p. 2).

This neglect is particularly problematic in the context of the United Nations 2030 Agenda for Sustainable Development, since, under SDG 8, the concept of decent work is coupled to economic growth as an explicit goal of sustainable development, with the aim to "create conditions for sustainable, inclusive and sustained economic growth, shared prosperity and decent work for all" [14], (p. 3). Target 8.9 specifically addresses tourism, calling for efforts to "devise and implement policies to promote sustainable tourism that creates jobs and promotes local culture and products" [51], (p. 99). Such an interpretation focuses on the quantity (i.e., the number of jobs created and employment multipliers) rather than the quality of individual employee's work. Bianchi and de Man [8] have recently complained that the UNWTO's indicators of sustainable tourism "reduce the interpretation of decent work merely to the 'number of jobs in tourism industries as a proportion of total jobs and growth rate of jobs, by sex" (p. 10). Overall, as Frey [49] notes in relation to SDG 8, "there is a conflation of the notion of 'business sustainability' with broader social aims of 'sustainable development', livelihoods and social and economic equity" (p. 1172). This conflation also applies to numerous interpretations of tourism and sustainability and, thus,

we strongly support the argument of many commentators who point out that rather than pursuing 'sustainable tourism' (understood primarily as sustaining tourism businesses) we should seek to identify what role individual tourism workers could play in a broader process of sustainable development [3].

In summary, debates about sustainability and tourism generally neglect themes related to tourism workers. Meanwhile, the handful of attempts to link labor issues with tourism sustainability, are dominated by a largely critical and pessimistic outlook, focusing heavily on the vulnerabilities of marginalized subjects of labor. We now turn to consider how to develop a fruitful research agenda that helps us link tourism work and workers within discussions pertaining to the social dimension of sustainable development.

### 3. An Agenda for Incorporating Tourism Work and Workers in the Sustainability Debate

Winchenbach et al. [19] offer a useful departure point for embedding tourism work and workers in the sustainability dialogue by arguing that focusing on decent work as per the ILO guidelines is problematic since it does not necessarily imply an improvement in the quality of the jobs. They contend that, if anything, the dominant neoliberal global environment leads to: "reduced job security; increase in humiliation and meaningless work; and lower pay and benefits" while also undermining "organisational functioning due to increased labor turnover, thus eroding an organisation's foundations for success" (p. 1029). Therefore, these authors propose that, when talking about jobs, the focus should shift towards building dignity and respect and overall improving the working conditions. They underline that although there have been policy moves at various levels to incorporate the issue of dignity while aiming to eliminate exploitation in the context of tourism, there is an obvious research gap on this topic.

These scholars [19] indicate that dignity in employment can be examined from the perspective of the "individual worker", the "organizational context" and "wider socioeconomic and policy context" (p. 1032). They believe that from each of the respective perspectives there are characteristics that can both promote but also violate dignity. For instance, when examining the "wider socioeconomic and policy context", a measure aimed at enhancing dignity would involve the adoption of a living wage law, while one that violates dignity would be the overriding tendency in a particular society to treat workers as a factor of production (see also [8]) rather than as individuals with varying levels of agency. Thus, establishing and maintaining dignity across the three levels becomes a contested practice regime [52], in which individual and institutional actors simultaneously generate, perform and adapt to the regulative, organizational and technological systems that frame dignified employment. In order, then, to establish an agenda that shifts tourism work and workers to the central stage of the sustainability debate, we propose that we must understand the dynamics of this practice regime as well as its distinct organizational and technological conditions while also deciphering the active role workers play in reproducing and disrupting it.

In the remainder of the paper, we sketch out two pertinent issues, which represent key challenges to creating dignity in the tourism labor market. These are: the aforementioned global neoliberal environment, which has led to the normalization of liminal and flexible workplaces in tourism; and technological (digital) transformations and new online platforms enabling self-employment and micro entrepreneurship. Subsequently, we will shed light on individual practices through which tourism workers actually shape their work and create meaningfulness and thriving in their jobs. By adopting the conceptual notion of job-crafting [53], we propose to treat workers as individuals with agency who are neither perpetual victims nor mere factors of production [23]. We illustrate the validity of our approach by addressing emerging organizational and job-crafting practices, in order to frame thriving at work in the so-called platform economy (also known as the collaborative economy, the sharing economy or the gig economy). Accordingly, a new research agenda for establishing the relationship between sustainability and the individual workers must address performances beyond relatively passive coping strategies while also revealing

how these individuals actively resist precarity in an organized fashion despite liminal, casualized and flexible work conditions.

### 3.1. Organizing Labor in Liminal Workplaces

Earlier we described the tendency in numerous studies of tourism sustainability to focus on the scale of individual destinations. We contended that this emphasis on local socio-ecological systems explains the relative absence of tourism workers in sustainability narratives, especially given these individuals' ambiguous and highly complex status when treated as part of the local community [18]. We suggest that the sustainability of tourism labor can only be understood in the context of broader processes of socio-economic restructuring including economic globalization and migration, the rise of neoliberal workfare policy regimes and the decline of collective representation and union membership. The concept of liminality captures the intersection of these larger scale trends with the experience of individual tourism workers. This concept was deployed by Underthun and Jordhus-Lier [54] to conceptualize both personal transitions and translocal positionality (for example in relation to labor migrants and working tourists) and also to suggest that flexible and/or precarious labor markets, increasing migration and widespread youth underemployment mean that liminality is emerging as a more structural societal characteristic.

Underthun and Jordhus-Lier [54] correctly assert that we should explore the experiences and motivations of different kinds of liminal tourism workers whose presence may undermine collective efforts to improve working conditions. A case in point is that of self-employed tourist guides, who voluntarily choose casual jobs below market rates, and might even agree to perform these without pay, for purposes such as pursuing their own passion for travel [55]. For young working tourists who are in a liminal position by choice, tourism work is a temporary and short-term activity undertaken to fund their travels or their studies. Thus, they may have little interest in workplace conditions. Other migrant groups, especially those dependent on a work permit, may resist collective efforts to improve working conditions because they fear losing their jobs altogether [56,57]. As Underthun and Jordhus-Lier [54] conclude, "the organisational presence of trade unions is perceived as static and rigid in the face of an increasingly fluid and flexible workforce" (p. 25). The research by McDowell et al. [58] on migrant workers at a London hotel and Rydzik and Anitha's [24] investigation of Central and Eastern European migrants working in tourism and hospitality in the UK have highlighted the general lack of collective action and engagement with trade unions. Even in a Scandinavian welfare state such as Norway, a survey of hospitality workers in Oslo showed that unionization rates were on average only 22% [59].

Nevertheless, the current situation, which can be seen as one that reinforces precarity, does not necessarily imply that tourism workers entirely lack agency in relation to maintaining dignity at work, or improving their working conditions. Rydzik and Anitha [24] identify resilience, reworking and resistance as the main strategies adopted by migrant tourism workers in response to employment-related exploitation. From these authors' perspective, resilience refers to day-to-day survival tactics to deal with oppressive conditions. They might, for instance, distance themselves from their work by breaking petty rules or refusing to put up with discriminatory acts. Meanwhile, reworking is a more active response, typified by changing jobs. This is geared towards improving an individual worker's conditions. Janta and Ladkin [60] and Lugosi, Janta and Wilczek [61], for example, have written about Polish hospitality workers in the UK, and how they use Internet discussion groups to find new jobs, while-warning their compatriots about unfair and abusive employers. Finally, resistance is the most radical of the strategies, since it seeks to transform the structures causing the precarious and oppressive working conditions in the first place. Thus, a key area for future research when it comes to the sustainability of tourism work and workers is to investigate how liminal tourism workers in different labor markets and workplaces maintain their dignity through resilience and reworking. A second area of interest is how labor organizations and trade unions can support liminal

workers in resisting and transforming neoliberal organizational structures, which, over the years, have accentuated and perfected flexibilization strategies, including temporal, functional and wage flexibility and outsourcing [62].

In addition to the seemingly inexorable expansion of neoliberal models, ongoing technological changes have been dramatically transforming employment structures in various industries, including the tourism and hospitality sectors. On the one hand, digital transformations, which have initiated various disruptive innovations including collaborative economy (peer-to-peer) platforms, offer opportunities for enhancing flexibility in the labor force but are, on the other hand, increasingly criticized for placing neoliberalism on "steroids" [63,64] and further aggravating already hyper-exploitative employment relations. At the same time, they open up new opportunities for resilience and reworking on the part of individual workers through job crafting [65]. We now briefly examine such digital transaction platforms and how they influence the provision, distribution and recruitment of labor. We particularly focus on how these transformations influence self-employment.

### 3.2. Technological Changes and the Enhancement of Flexible Self-Employment

Interactive digital technologies have enhanced old, while creating novel, forms of exchanges between people and businesses. So-called platform business models that host such peer-to-peer transactions (buying, selling, renting, sharing, swapping) not only disrupt the way we circulate goods, information and property related to travel and tourism, but also affect established employment structures. Short-term accommodation rental [66], home swapping [67], free walking tours [68], ridesharing [69] and dinner-sharing offer casualized working conditions through the alluring rhetoric of empowerment and flexibility. Slogans such as "enjoy the liberty of being your own boss" (Copenhagen walking tours), "take the leap and quit your 9-to-5 job" and "be a part of a community of millions of entrepreneurs" (Airbnb) promote an entrepreneurial ethos that highlights the benefits of self-employment against the constraints of contracted jobs. These shifts have led to the appearance of a new generation of tourism workers, including travel bloggers, influencers, Airbnb hosts, and Uber drivers, all of whom witness working conditions and existential security, which are almost never safeguarded by clear national legislation or guidelines by the International Labor Organization. As such, platform labor is concurrently flexible and precarious. In fact, several authors, including Van Doorn et al. [27], have commented on the vulnerability of platform laborers in the gig economy.

Researchers from different disciplines have addressed both the opportunities [70] but also the dark side of digital microentrepreneurship. In her seminal critique of surveillance capitalism, Shosanna Zuboff [71,72] warns against the long-term costs of the commodification and datafication of human lives. This also applies to flexible platform workers, who are algorithmically recruited, monitored and rewarded to maintain high levels of service quality provided by self-employed staff. For instance, through a communitarian ideology, Airbnb constantly encourages its "superhosts" to engage in self-disciplining and self-regulative practices by the platform's automated tracking and feedback features [66]. Thus, rather than being one's own boss, platform workers are surveilled by artificial intelligence (AI) that creates new, cunning forms of economic, physical and emotional vulnerabilities [66].

Increasing flexibility, but also enhanced levels of precarity dominate various subsectors of tourism and hospitality services including some of the very largest players (e.g., transnational hotel chains, cruise lines, major tour operators and guiding services). Every one of these sectors displays levels of division of labor reflected along ethnic, racial, (dis)ability and gender lines [73–75]. The lack of intra-organizational measures to protect an increasingly flexible workforce as well as the inability of labor unions to embrace these groups further accentuates the liminality of many tourism workers. It seems, therefore, that despite the positive rhetoric around the freedom and flexibility afforded by digital platforms, they in fact recreate existing patterns of exploitation. Yet, as we have already mentioned, despite lacking an institutionalized safety net, these workers are able to produce novel social networks and work-related communities, and ultimately foster dignity beyond

conventional structures. This leads us to underline the importance of job crafting [53] but also other autonomous capacity-building mechanisms, which workers adopt in order to thrive in their job and maintain or even enhance their dignity.

### 3.3. Job Crafting and Other Proactive Labor Practices

Job crafting refers to mundane practices through which individuals create personal meaningfulness, fulfil interests, and play up to individual strengths in their work. It is a proactive and self-initiated behavior, in which employees alter (design or craft) their work roles by changing specific work tasks, thoughts and perceptions about work, and work relationships [65]. Thus, job crafting is instrumental in terms of generating "positive outcomes, including engagement, job satisfaction, resilience and thriving" at work [53]. It can entail three practices, which are: redesigning task boundaries such as changing job tasks in volume or form; reworking cognitive task boundaries (changing their own or others' view of the job); and changing relational boundaries to other actors. We briefly illustrate how task-related job crafting, cognitive job crafting, and relational job crafting resonate with resilient practices in tourism.

When engaged in job crafting, individuals adopt practices to transform their work-related tasks and might volunteer to embrace responsibilities better suited to their interests and strengths. They might also redesign how tasks are meant to be accomplished [63]. Workers' flexibility for task crafting may greatly differ and can be conditioned by job descriptions, motivation, compensatory mechanisms and career opportunities. Importantly, an individual's life situation also plays a role in determining the level of flexibility one has in terms of task crafting. Mobile creative workers such as seasonal guides [76] or digital nomads [70] happily accept precarious and insecure conditions if they can integrate their passion or leisure interests with their jobs. Extreme sport entrepreneurs are often more than ready to alter facets of their work (e.g., expanding or minimizing certain activities) in order to fully engage with their hobbies over a longer period [77]. The landscape of agency is variegated, and it has been noted [78] that the job satisfaction and perceptions of vulnerability of those for whom the gig represents supplemental income greatly differs from those who existentially depend on it.

However, task crafting is not only limited to creative jobs. Even self-initiated, micro-level changes in highly standardized contexts may give employees the feeling of being in control and of thriving in their present job [53]. A study in Danish supermarkets revealed that cashiers who were given the opportunity to propose incremental innovations to improve uniform checkout operations considerably enhanced their self-image and felt a better connection to their fellow colleagues [79]. Regardless of their scale, such practices provide workers with a sense of accomplishment and meaning, which is related to cognitive or perception crafting. As Rydzik and Anitha [24] have indicated, in the case of UK housekeeping staff, these workers constantly challenge the cognitive boundaries of their work, by shifting how they think about and process job-related experiences. Some choose to focus on the most rewarding or fulfilling moments at work, whatever these may be, while others search for meaningfulness by strengthening connections between work and private life situations. The reflexive loops related to perception crafting can not only build stronger personal identities, but also set into motion practices of resistance or reworking [18]. Finally, job crafting relates to changing the relational boundaries of work to derive meaningfulness and identity [65]. People may create new working relationships or reframe the purpose of existing ones, for instance, by building alliances or becoming mentors. This aspect of job crafting is especially obvious in cases where individuals create their own safety net in liminal working conditions. For instance, tourism workers are often connected to various informal networks (e.g., recruitment pipelines, social ties, and even virtual communities) all of which provide new avenues for organized resistance. Examples of this connectivity are provided by Facebook groups driven by Airbnb superhosts. The website Glassdoor offers a platform for current and former employees to anonymously review their companies and compare salary levels. In a larger study of African gig workers,

Anwar and Graham [80] (p. 1278) demonstrate that platform workers engage in remarkably diverse agency practices to re-seize control, including running online training classes and advising fellow workers, exposing and filtering bad clients or simply operating with multiple accounts.

Thus, although neoliberal globalization has arguably enhanced precarity and normalized liminality, the digital networks, which support globalization, also enable the development of virtual communities and possibilities for transnational, self-organizing and grassroots labor movements. Future research endeavors should place more focus on mapping and assessing these new dynamics of organized labor and also assess how the generation of such new virtual/footloose labor communities complement institutionalized labor protection and safety mechanisms.

## 4. Conclusions: Towards A Human-Centered Agenda for Sustainable Tourism Employment

Most debates about tourism and sustainability focus on the possibility of reconciling economic growth and environmental protection. This largely derives from the neoliberal "discourse of tourism as an industry" [3], (p. 1192), and the contradictory logic that dominates discussions about tourism and sustainability, which on the one hand argues for perpetual growth while, on the other hand, seeks to protect the environment in a spirit of equity and social justice [8]. Within this narrative, the social aspects of sustainability, especially the role of work and individual workers, are often hidden [25], despite the widespread recognition that precarity dominates in the tourism labor market.

A major obstacle to moving the discussion forward is the tendency in both the resource and the community-based approaches to sustainability to emphasize local aspects or elements (residents, resources, and culture) of socio-ecological systems. This emphasis on native/indigenous conditions renders it hard to embrace tourism workers who tend to be highly mobile and have ambiguous or temporary ties to the communities in which they work [18]. Consequently, they are often treated in an aggregated manner as objectified labor power, for instance, in terms of their contribution to the employment multiplier or a reduction in unemployment. As Bianchi and de Man [8] correctly surmise, the problem with treating workers as statistics and mostly as a factor of production is that it diminishes these individuals' humanity. Unfortunately, during the Anthropocene, this objectification of workers as passive actors without contingent agencies dovetails with Wakefield's [9] argument that people are regularly seen as separate from the environment in which they exist. This, in final analysis, goes a long way to explain why the issue of tourism work and workers is regularly obfuscated in discussions concerning tourism and sustainability.

Inspired, among others, by Winchenbach et al. [19], we have argued that future research should integrate work and workers in the tourism sustainability debate by focusing on dignity and the possibilities for transforming working conditions and creating decent work. Focusing on the quality of paid employment and labor market relations—rather than accepting neoliberal interpretations of economic sustainability that focus on the number of jobs created—opens up space for a wider debate about the meaning of sustainable tourism. We have indicated that, inter alia, the pursuit of this objective is conditioned by the increasingly liminal and flexible workplaces in the tourism industry but also through the rapid transformations driven through digital transformations, including the appearance of platform-based tourism products. With this backdrop in mind, we have argued that even though the degree of institutional protection of workers (e.g., though unionization and welfare policies) is waning, certain individuals are becoming increasingly adept in making their own decisions, which shape their job in a manner that increases its meaningfulness. The concept of job-crafting as it relates to the collaborative economy underlines the significance of human agency. Specifically, it opens the door for us to treat workers as resourceful individuals with agency who are neither mere factors of production nor perpetual victims. Through job crafting and other practices of resilience, reworking and resistance, tourism workers themselves may contribute to the practice regime of decent work. They can, in a number of instances, seek to change what they do on the job from day to day or alter their

idea about what the job means to them while aiming to develop new relationships with workers, employers and other actors with a view to transforming the structural conditions of precarity. We accept, of course, that not all workers have the same ability in terms of job crafting and that, at the end of the day, there are millions of individuals whose precarity is so extreme that their agency is severely undermined. This has especially been the case during the COVID-19 pandemic, which has devastated the tourism industry in many parts of the world and has left precarious workers stranded in an impossible situation [81,82]. Nevertheless, we believe that our approach in this paper offers a perspective that helps us move on from the extremely pessimistic "no hope" discourse that we often encounter on this topic towards a fine-tuned understanding of labor sustainability. Specifically, our approach, which along the lines of Rydzik and Anitha [24] "offers a more differentiated account of agency that resists both binary constructions of victimhood and agency as well as simplistic celebrations of unmitigated resistance" (p. 896), can be a promising launching point for strengthening our understanding of how work and workers fit into the broader tourism sustainability narrative. Such an endeavor must set off by exploring the variegated landscape of power, agency and self-organization among casualized and precarious workers. Instead of focusing on single segments of the labor market, we must explore the connections and regulative similarities between conventional and platform-operated tourism employment. There is a need to better understand the dynamics of the emerging, casualized labor market of tourism (e.g., labor force mobility, new hierarchies) and its consequences for community cohesion, social integration and new dependencies.

Finally, we strongly believe that an approach such as the one we have advocated opens up avenues, which will allow researchers to escape the neoliberal manner in which the SDGs tend to be treated. Hopefully, this will enable us to elevate the treatment of the social equity dimension of tourism sustainability to the same status as that of environmental and economic growth concerns.

**Author Contributions:** D.I.: Conceptualization; Literature review, Writing—original draft; Writing—review & editing. S.G.: Conceptualization; Writing—review & editing. L.J.: Conceptualization; Writing—review & editing. All authors have read and agreed to the published version of the manuscript.

**Funding:** This research received no external research funding.

**Institutional Review Board Statement:** Not applicable for studies not involving humans.

**Informed Consent Statement:** Not applicable for studies not involving humans.

**Data Availability Statement:** No new data were created or analyzed in this study. Data sharing is not applicable to this article.

**Conflicts of Interest:** The authors declare no conflict of interest.

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
