# Peer review of "From Liminal Labor to Decent Work: A Human-Centered Perspective on Sustainable Tourism Employment"

_sustainability, doi:10.3390/su13020851_

Round 1

Reviewer 1 Report

The topic is very interesting, and it's very well presented. Nowadays, the current paper seems to be a theoretical introduction for an empirical work. In my opinion, authors have to demonstrate that they have done a systematic literature review in order to consider this theoretical paper as a quality paper for publishing. Authors have to explain the followed methodology in order to apply the sistematic literature review. Some references about this methodology can be found in usual data bases. If they didn't do a sistematic review, they can add the information obtained from the application of the methodology to their conclusions.

Author Response

A detailed response to your comment is provided in the attached file. Based on the comments of the other reviewers and the response of the guest editor of this issue we have decided not to make the change which you suggest since this would mean entirely rewriting the paper and would shift its focus, which is essentially to be a conceptual paper for a special issue. We thank you for your very good comment but we will have to take it into consideration for another paper in the future.

Reviewer 2 Report

This is a key issue, well done for contributing to the discourse

there is a general neglect re tourism employment (see Ladkin 2011 & Baum et al 2016[taxonomy] even before considering sustainabilty arguments and this may be worth registering too

here are two papers I am aware that speak almost directly to the gap you suggest you redress the neglect of - while you cite both (Winchenbach et al & Robinson et al) and there may be others - you could tone down the claim?

A JTR paper (Robinson et al 2014 [core-periphery] directly makes the point about labour not being where it is needed - might be worth referring to

yes Baum can be a bit dour, but AI/robotisation may also improve quality of work? Wirtz, J., Patterson, P. G., Kunz, W. H., Gruber, T., Lu, V. N., Paluch, S., & Martins, A. (2018). Brave new world: service robots in the frontline. Journal of Service Management.

re choices and precarious work there is some nascent literature on food delivery gig workers perhaps, that fleshes out the tension between precarity and agency?

in discussion of migrants and women is intersectionality worth a guernsey?

the point you make ending [35] i think is worth developing a bit more, given your goal of providing a balanced perspective

i think the point you make re external ownership transferring to a loss of voice at community level is also worth some development

re migrants and agency and negotiation might Hania Janta's work be useful?

i haven't developed this much but as you mention disruptors, like AirBnB (mostly investors contrary to side-hustle narratives that Dolnicar senselessly peddles) and other gigs, they too are a double-edged sword - they are precarious and offer choice to those on the fringes but esp in the case of Airbnb steal demand (and so work) away from the formal economy

job crafting can be deconstructive too - see Lugosi recently on deviance and Robinson (2008) re chefs - also Lynch re humour

really enjoyed the paper, and the idea of seeking a constructive way forward - good luck

Author Response

Thanks for your very insightful comments. We have attached a detailed table that indicates how we have responded to each of your comments. Hopefully, this is to your satisfaction.

Reviewer 3 Report

I have only minor comments (see below) for this article as it is a thought-provoking, well-written and timely paper that is - more or less - ready for publication as it is. In saying that, I wonder if the authors could perhaps be more critical in their conclusions? I think they have done what they set out to do – suggesting a research agenda that moves us beyond staid and pessimistic discourses. However, I wonder if it is worth considering how these potential avenues for research could allow for active engagement/resistance/contestation/opposition of current neoliberal interpretations of the SDGs and tourism’s biased focus on the environmental and economic elements?

Below are a few other comments the authors may want to consider:

Page 1 line 32: I think this first paragraph works very well. My only question revolves around the sentence where the authors say ‘While stressing that the most vulnerable persons…’.  I wondered if this sentence was supposed to highlight the groups whose are most vulnerable to precarious employment or whether the intention was to say something more about these groups. I think I just read the sentence like it was supposed to say more – the wording ‘whilst stressing that…’ just suggested more than a list of these very vulnerable groups.

Page 2 final two paragraph: I fully agree with the authors that we need to think about the heterogeneity of tourism work and tourism workers and that we need to find ways to ensure that tourism workers are not perceived or assumed to be voiceless. However, I think that these last two paragraphs immediately point to some of the challenges, tensions and conundrums we face in discussing tourism work. I would ask the authors to re-look at these two paragraphs and consider how to reconcile the hope of developing a new research agenda and positioning tourism workers in the sustainably debate fit together. For me, at the moment, these two paragraphs read a little like the authors are trying to do two quite separate and distinct things in this paper. I do not think that this is what they intended. How can the author’s re-phrase/re-frame to show how the two paragraphs lead to this future research agenda? 

Page 3 paragraph starting line 119: The middle part of the paragraph is a bit ‘clunky’ and does not really fit together. The middle sentence about jobs being negatively compared to those in other sectors does not seem to really link to the previous sentence nor does it directly link into the sentences after it which talk about specific groups of workers and specific vulnerabilities but not compared to other sectors.

Page 4 line 156, first sentence: I think the ‘or’ here is supposed to be ‘of’. In the next sentence, I think the word ‘especially’ is redundant and could be deleted.

Page 8 lines355-356: I think there should be either an ‘and’ or an ‘or’ between McDowell’s study and Rydzik and Anitha’s investigation?

Page 9 line 415: and I think along (dis)ability lines as well.

Page 10 line 453: I am not sure which example the authors are referring to? This is the first and only mention of housekeeping staff int eh article. Are the authors referring to Rydzik and Anitha’s study on page 8?

Page 10 line 475: I think ‘derives’ should be ‘derived’?

Author Response

Thanks for your very insightful comments. We have now made the changes you have suggested and we have addressed them as seen in the attached table.
